# Isoorientin Attenuated the Pyroptotic Hepatocyte Damage Induced by Benzo[a]pyrene via ROS/NF-κB/NLRP3/Caspase-1 Signaling Pathway

**DOI:** 10.3390/antiox10081275

**Published:** 2021-08-11

**Authors:** Hao Li, Li Yuan, Xueyi Li, Ying Luo, Zhong Zhang, Jianke Li

**Affiliations:** Engineering Research Center of High Value Utilization of Western China Fruit Resources, College of Food Engineering and Nutritional Science, Shaanxi Normal University, Xi’an 710119, China; lihr13@snnu.edu.cn (H.L.); lixueyi@snnu.edu.cn (X.L.); luoying@snnu.edu.cn (Y.L.); zzhang@snnu.edu.cn (Z.Z.); jiankel@snnu.edu.cn (J.L.)

**Keywords:** isoorientin, benzo[a]pyrene, pyroptosis, ROS, NF-κB, NLRP3

## Abstract

Isoorientin (Iso), a natural bioactive flavonoid, possesses significant anti-tumor and anti-oxidant activities. Benzo[a]pyrene (BaP) is a food processing injurant with carcinogenicity, teratogenicity, and genotoxicity. Our preliminary study demonstrates that Iso attenuated the pyroptotic hepatocyte damage induced by BaP; however, the molecular mechanism remains unknown. The present study showed that Iso reduced the increase caused by BaP in the overflow of LDH, NO, and the electrical conductivity and the protein expressions of GSDMD-N, IL-18, and IL-1β, further showing that Iso could reduced the pyroptotic damage in HL-7702 cells induced by BaP. Caspase-1 inhibitor (Z-VAD-FMK) inhibited the characteristic pyroptosis protein expressions of Caspase-1, GSDMD-N, IL-18, and IL-1β, showing that the classic pyroptosis pathway depending on Caspase-1 was caused by BaP in HL-7702 cells. Consistent with the effects of the NLRP3 inhibitor (MCC950), NF-κB inhibitor (PDTC), ROS, and mtROS inhibitor (NAC and Mito-TEMPO), Iso weakened the stimulatory effects of BaP on the levels of ROS, the nuclear localization of NF-κB, and the activation of NLRP3 inflammasome and the characteristic indices of pyroptosis, demonstrating that Iso could alleviate the BaP-induced pyroptotic hepatocytes injury through inhibiting the ROS/NF-κB/NLRP3/Caspase-1 signaling pathway, which provides a new perspective and strategy to prevent liver injury induced by BaP.

## 1. Introduction

Isoorientin (3′,4′, 5, 7-tetrahydroxy-6-C-glucopyranosy flavone, Iso, Figure 1A), a natural flavonoid compound, is abundant in food such as hawthorn [1], cucumber [2], and pueraria lobata [3], with anti-oxidation, anti-inflammation, and anti-tumor activity [4,5]. It is able to prevent liver injury by reducing lipid metabolism, enhancing the ability of antioxidation, regulating the secretion of pro-inflammatory cytokines [6], and suppressing the enzyme activities of the respiratory chain complexes and the phase II [7].

Benzo[a]pyrene (BaP, Figure 1B), is a group I carcinogen and possesses strong carcinogenicity, teratogenicity, and genotoxicity [8,9,10]. It is widespread in the high temperature heat processed foods, such as smoked fish, sausage, fried chicken, vegetable oil, and cereals [11]. Pyroptosis is an inflammatory programmed cells death, accompanied by the formation of pores on the cell membrane and the release of contents, which in turn triggers an inflammatory response [12]. It has been reported that Gossypol [13], Arsenic [14], and Cadmium [15] induced the pyroptotic damage.

NLRP3 (NLR family pyrin domain-containing 3) inflammasome is a multiprotein complex, formed by NLRP3, apoptosis associated speck-like protein containing a CARD (ASC), and Pro-Caspase-1. It could be expressed in large quantities, when inflammation occurs in hepatocytes [16]. Existing studies have proven that NLRP3 inflammasome plays a critical role in the process of pyroptosis [17,18], it induces pyroptosis by promoting the cleavage and activation of Caspase-1, and resulting in the maturation and secretion of pro-inflammatory cytokines [19]. NLRP3 inflammasome could be activated by the extra stimuli, the mitochondrial dysfunction, reactive oxygen species (ROS), lysosome damage, or disorder of intracellular ion homeostasis [20,21,22]. Besides, the activation of NLRP3 inflammasome requires the stimulation of nuclear factor-κb (NF-κB) [23]. NF-κΒ is the key to the biological processes of inflammation, proliferation, differentiation, and it controls the transcription of pro-inflammatory genes that regulate the synthesis of chemokines, cytokines, and adhesion molecules [24].

Our preliminary study demonstrates that BaP induced pyroptotic liver injury through improving the electrical conductivity, stimulating the release of LDH and the production of NO, up-regulating the protein expression of pyroptosis characteristic indices, such as Caspase-1, COX-2, IL-1β, and IL-18 in vitro and vivo [25], and Iso could significantly attenuate the pyroptotic hepatocytes damage induced by BaP [26]; however, the underlying mechanism remains unknown.

This study aimed to explore whether or not the natural flavonoid Iso attenuated BaP-induced pyroptotic hepatocyte damage through the ROS/NF-κΒ/NLRP3/Caspase-1 signaling pathway in HL-7702 liver cells by detecting ROS levels, the nuclear localization of NF-κΒ, the activation of NLRP3 inflammasome, and the characteristic protein expressions of pyroptosis.

## 2. Materials and Methods

### 2.1. Chemicals and Reagents

Iso (purity ≥ 98%) was obtained from Pharmaceutical Technology Co., Ltd. (Jiangsu, China) and BaP was purchased from Sigma (St. Louis, MO, USA). Caspase-1 activity assay kit and 2,7-dichlorodihydrofluorescein diacetate (H_2_DCFDA) were obtained from Beyotime biotechnology Co., Ltd. (Shanghai, China). RPMI-1640 medium, BCA protein kit, and penicillin–streptomycin solution were from Thermo Fisher (Shanghai, China).

TritonX-100, DAPI-Fluoromount-G, Bovine Serum Albumin (BSA), and CoraLite488-conjugated Affinipure Goat Anti-Rabbit IgG(H+L) were from Proteintech Group, Inc (Rosemont, IL, USA).

Polyclonal antibodies specific to NLRP3 (GTX106313), ASC (10500-1-AP), NF-κΒ (10745-1-AP), GSDMD (20770-1-AP), IL-1β (16806-1-AP) and IL-18 (SC-6179) were purchased from Santa Cruz Biotechnology (Dallas, Texas, United States). GAPDH (BA2913) was obtained from Bioworld Technology, Inc. (Louis Park, MN, USA). Caspase-1 (2225) was purchased from Cell Signaling Technology (Shanghai, China).

Specific inhibitors MCC950, N-acetylcysteine (NAC), and Z-YVAD-FMK were obtained from Beyotime biotechnology Co., Ltd. (Shanghai, China). Ammonium pyrrolidinedithiocarbamate (PDTC) and Mito-TEMPO were obtained from Sigma (St. Louis, MO, USA). All other reagents were dissolved in water or DMSO, and then diluted with fresh RPMI-1640 medium.

### 2.2. Cell Culture

HL-7702 liver cells were from the Shanghai Zhong Qiao Xin Zhou Biotechnology Co., Ltd. (Shanghai, China). Cells were cultured in RPMI-1640 medium with 10% fetal bovine serum and 1% penicillin–streptomycin, in a humidified incubator (5% CO_2_, 95% air) at 37 °C.

### 2.3. Cytotoxicity Assay

Cell viability was detected by the MTT assay. HL-7702 cells with the density of 1 × 10^6^ cells/mL were seeded into plates. After treatment, the MTT (0.5 mg/mL) was added and incubated for 4 h. Then, MTT was removed and 100 μL DMSO was added to each well. The absorbance at 490 nm was detected using a microplate reader (Thermo Fisher, Ltd., USA). Cell viability of HL-7702 cells was expressed as the ratio of the absorbance of the treated group cells to the control group cells.

### 2.4. Cell Cycle Analysis

Cell cycle was examined by Flow cytometry. After treatment for 24 h, trypsin was added to digest and blow the cells to prepare a cell suspension. Then, they were centrifuged and resuspended several times. Finally, RNase A and PI were added to the resuspended cells which were then cultured for 30 min at room temperature in the dark, and analyzed by the Flow cytometry (Partec GmbH, Münster, Germany).

### 2.5. Measurement of Intracellular ROS

ROS was detected using Flow cytometer and Fluorescence microscopy was used to detect ROS levels. Cells were harvested and incubated with 10 µM H_2_DCFDA, which was added to the treated cells for 20 min at 37 °C. After washing with cold PBS, the fluorescence of cells was observed under Fluorescence microscope (Olympus Optical, Tokyo, Japan) or Flow cytometer (Partec GmbH, Münster, Germany).

### 2.6. Western Blot Analysis

HL-7702 cells were lysed with a lysis solution containing RIPA buffer, and then the protein concentration of the supernatant after centrifugation was measured with BCA kits, and finally, the protein concentration of all treatment groups were balanced with a loading buffer, and denatured in a 95 °C water bath for 10 min. Subsequently, the treated proteins were subjected to such steps as loading, electrophoresis, and transfer to a polyvinylidene fluoride membrane, followed by washing, blocking with skimmed milk powder for 2 h, and incubation with the primary antibody for 4 °C overnight and the secondary antibody for 2 h. Finally, the ECL kit was used for coloration and proteins were visualized on the Chemical XRS Imaging System (UVP, Ltd., USA).

### 2.7. Release of LDH

The contents of Lactate dehydrogenase (LDH) in the cell-free supernatant reflect the cellular permeability. The absorbance value of the cell-free supernatant at 450 nm was determined by using an LDH assay kit (Nanjing Jiancheng Bioengineering Institute, Nanjing, China).

### 2.8. NO Assay

Griess reagent (including 0.1% (*w*/*v*) N-(1-naphathyl)-ethylenediamine and 1% (*w*/*v*) sulfanilamide) was used to determine the content of NO in the culture supernatant. After treatment, the cell-free supernatant was collected and reacted with Griess reagent at 25 °C for 15 min. The absorbance at 540 nm was detected by the microplate reader (Thermo Fisher, Ltd.,USA). The value of NO was calculated using NaNO_2_ standard curve.

### 2.9. Relative Electrical Conductivity Assay

The relative electrical conductivity of the cell-free supernatant was measured by the DDSJ-308A conductivity meter (Shanghai precision scientific instrument co., Ltd., Shanghai, China).

### 2.10. Scanning Electron Microscopy

Cells were fixed with 2.5% glutaraldehyde, and were sequentially dehydrated in graded ethanol (30, 50, 70, 90, 100% (*v*/*v*)), ethanol:isoamyl acetate (3:1, 1:1, 1:3 (*v*/*v*)), and isopropyl acetate for 15 min at room temperature. Finally, cells were observed under a scanning electron microscope (HITACHI, SU8220, Tokyo, Japan) at a voltage of 5 kV.

### 2.11. Caspase-1 Enzyme Measurements

The activity of Caspase-1 enzyme in cells was detected by a Caspase-1 enzyme kit (Beyotime Biotechnology, Shanghai, China). Briefly, cells lysates containing 10–30 mg of protein reacted with the reaction buffer containing Ac-YVAD-ρNA at 37 °C. After 30 min, the absorbance at 405 nm was detected using a microplate reader (Thermo Fisher, Ltd., USA.) and the Caspase-1 activity was normalized for total proteins of cell lysates.

### 2.12. NF-κB Activation Assay

Firstly, HL-7702 cells were added to the cell coverslips for 24 h. After fixing (ice-cold methanol), permeabilizing (0.25% Trition X-100), and sealing, cells were incubated with NF-κB antibody (1:100) and FITC-conjugated antibody (1:200) for 2 h and 1 h, respectively. Finally, cells were stained with DAPI for one minute, and then observed with an Olympus FV1200 confocal microscope.

### 2.13. Statistical Analysis

All experimental results are expressed as mean ± standard deviation (SD) and were analyzed by ANOVA and Duncan test (SPSS v19.0); *p* value < 0.05 was considered as statistically significant.

## 3. Results

### 3.1. Effects of BaP on the Proliferation of HL-7702 Cells

MTT assay was used to measure cell viability. BaP induced the death of HL-7702 cells in a dose-dependent manner at three concentrations of 10 µM, 25 µM, and 50 µM (*p* < 0.01). With the increasing concentration of BaP, the cell viability decreased by 35.73%, 38.24%, and 42.15%, respectively (Figure 1C). There were significant differences among these groups (*p* < 0.01). With microscope observation, we also found that the degree of cell damage was positively correlated with the concentration of BaP (Figure 1D). In order to explore whether the inhibitive effects of BaP on cell proliferation is related to blocking the cell cycle, the flow cytometry was calculated. As shown in Figure 1E, compared with the control, BaP significantly (*p* < 0.01) prolonged the S-phase by 12.46% and shortened the G_2_-M-phase by 10.78%. These results indicated that BaP not only could damage HL-7702 cells, but also suppress the proliferation of HL-7702 cells by blocking the S-phase.

### 3.2. BaP Induced Pyroptotic Damage in HL-7702 Cells

Pyroptosis is a form of the programmed cell death which accompanies inflammation. When pyroptosis occurs, the GSDMD-N is activated to oligomerize on the cell membrane and form pores, which leads to the change of cell membrane permeability. As shown in Figure 2E, there were few pores in the control group, and the number of pores increased with the increase of the concentration of BaP. Furthermore, compared with 0 µM BaP treatment, 50 µM BaP notably (*p* < 0.01) raised the overflow of LDH (Figure 2A) and NO (Figure 2B), and the electrical conductivity (Figure 2C) by 57.21%, 88.86%, and 87.17%, respectively, and resulted in the significant increasing trend in the protein expression of GSDMD-N (Figure 2D). These results indicated that BaP could induce the pyroptotic hepatocyte damage in HL-7702 cells.

### 3.3. BaP Induced Caspase-1-Dependent Pyroptotic Hepatocytes Damage

The classic pathway of pyroptosis is dominated by Caspase-1. Z-VAD-FMK is an irreversible Pan-Caspase inhibitor that can inhibit the activity of the Caspase family. We chose Z-VAD-FMK to verify whether or not the classic pathway of pyroptosis has been induced by BaP in HL-7702 cells. As shown in Figure 3A, in comparison with the control group, 25 µM BaP obviously (*p* < 0.01) decreased the cell viability by 39.40%, while the viability of cells which were exposed by Z-VAD-FMK and BaP only decreased by 7.3%, comparing with the control group. The same trends were found in the overflow of LDH (Figure 3B), NO (Figure 3C), and the electrical conductivity (Figure 3D). Similarly, Z-VAD-FMK weakened the enhancing effects of BaP on the Caspase-1 enzyme activity (Figure 3E), the protein expressions of Caspase-1, GSDMD-N, and inflammatory factors (IL-18, IL-1β) (Figure 3F), showing that BaP induced Caspase-1-dependent pyroptotic hepatocyte damage in HL-7702 liver cells.

### 3.4. Effects of NLRP3 Inflammasome on BaP-Induced Pyroptotic Hepatocyte Damage

We assessed whether or not the activation of NLRP3 inflammasome is related to the occurrence of pyroptosis. MCC950 is an effective and selective inhibitor of NLRP3. It was obvious from Figure 4A that the addition of MCC950 could greatly (*p* < 0.01) reduce the degree of cell damage and improve cell viability. In the co-treatment with MCC950 and BaP group, LDH release (Figure 4B), NO (Figure 4C), and the electrical conductivity (Figure 4D) were significantly (*p* < 0.01) lower than in the BaP group. Moreover, the characteristic protein expressions of pyroptosis (Figure 4F) were also reduced by MCC950, as well as the Caspase-1 enzyme activity (Figure 4E), contrary to the result of the BaP group. These results fully proved that BaP induced the pyroptotic hepatocyte damage through activating NLRP3 inflammasome in HL-7702 liver cells.

### 3.5. Effects of NF-κB on NLRP3 and Pyroptotic Hepatocytes Damage

Inhibition of NF-κΒ can effectively reduce the level of pro-inflammatory cytokines. PDTC is an inhibitor of NF-κB activation that can inhibit the nuclear localization of NF-κB in cells. To explore whether or not BaP induced Caspase-1-dependent pyroptosis through activating NF-κB, then stimulating NLRP3 inflammasome, cells were exposed to PDTC. As shown in Figure 5A, PDTC could significantly (*p* < 0.01) alleviate the decline of cell viability caused by BaP. In comparison with the BaP treatment, 50 µM PDTC would significantly (*p* < 0.01) weaken the increase of LDH release (Figure 5B), NO (Figure 5C), and the electrical conductivity (Figure 5D) caused by BaP. The protein expressions of NF-κΒ, NLRP3, ASC, Caspase-1, GSDMD-N, IL-18, and IL-1β in the BaP and PDTC co-treatment group were also lower than those of the BaP group (Figure 5E). Meanwhile, BaP resulted in an obvious nuclear localization of NF-κB (p65), while PDTC could suppress the nuclear localization of NF-κΒ (p65) (Figure 5F). From the above, these results indicated that BaP induced Caspase-1-dependent pyroptosis through activating NF-κΒ, and then stimulating NLRP3.

### 3.6. Effects of ROS on NF-κB, NLRP3, and Pyroptotic Hepatocyte Damage

BaP exposure would generate a large amount of ROS, and ROS could activate the nuclear localization of NF-κΒ and NLRP3 inflammasome. NAC is a common antioxidant, which can remove ROS in cells. Mito-TEMPO was proven to specifically remove ROS in mitochondria. In order to understand what role ROS plays in the pyroptotic hepatocyte damage caused by BaP, NAC and Mito-TEMPO were used to treat the HL-7720 cells. It was shown that NAC or Mito-TEMPO significantly (*p* < 0.01) improved the viability of cells after BaP exposure (Figure 6A) and obviously slowed down the production of ROS (Figure 6B). Our results also showed that the addition of NAC or Mito-TEMPO would greatly (*p* < 0.01) decrease the overflow of LDH (Figure 6C), NO (Figure 6D), and the electrical conductivity (Figure 6E). In addition, the protein expressions of NF-κΒ, NLRP3, ASC, Caspase-1, GSDMD-N, IL-18, and IL-1β in the BaP and NAC or Mito-TEMPO co-treatment group were lower than those of the BaP group (Figure 6F). According to the previous results, it was concluded that BaP induced the pyroptotic hepatocyte damage in HL-7702 cells by activating the ROS/NF-κΒ/NLRP3/Caspase-1 signaling pathway.

### 3.7. Effects of Iso on the BaP-Induced Pyroptotic Hepatocyte Damage

Our previous work proved that Iso inhibited the pyroptosis induced by BaP; this study further verified whether or not Iso inhibited the pyroptotic hepatocytes damage induced by BaP in HL-7702 cells through the ROS/NF-κΒ/NLRP3/Caspase-1 signaling pathway. Firstly, we carried out the assay of MTT and cell cycle to evaluate the effects of Iso on HL-7720 cells proliferation. The viability of cells decreased in the BaP group, but increased in the Iso and BaP co-treatment group (Figure 7A). Compared with the BaP group, the percentage of S-phase in the cell cycle was decreased by 3.88% after Iso and BaP co-exposure (Figure 7B). At the same time, Iso decreased the levels of ROS (Figure 7C) and the Caspase-1 enzyme activity (Figure 7D), as well as the characteristic protein expressions of pyroptosis (Figure 7F). Besides, in comparison with the BaP treatment, 5 µM Iso reduced the number of cell membrane pores (Figure 7E) and significantly suppressed the nuclear localization of NF-κΒ (p65) (Figure 7G). All the results allowed to conclude that Iso had an alleviative effect on the BaP-induced pyroptotic hepatocyte damage in HL-7702 cells through the inactivation of the ROS/NF-κB/NLRP3/Caspase-1 signaling pathway.

## 4. Discussion

Pyroptosis is a programmed cell death mode which depends on the inflammatory Caspase family and is accompanied by an inflammatory reaction. Long-term and excessive pyroptosis will lead to cell death, tissue damage, organ failure, septic shock, and other pathological conditions [27]. It has been reported that Gossypol, Arsenic, and Cadmium induced the pyroptotic damage [13,14,15]. Our preliminary study also demonstrates that BaP caused hepatocytes damage via inducing pyroptosis [25,28]. Iso is a natural flavonoid with a luteolin structure that is widespread in some foods and edible plants [29,30]. It could improve cell viability, reduce the electrical conductivity, NO release, and the protein expression of Caspase-1 to alleviate BaP-induced pyroptotic liver injury [26]. In this study, we further verified that Iso could reduce the membrane pores, the nuclear localization of the p65 subunit of NF-κΒ, the activation of NLRP3 inflammasome, the increase of Caspase-1 enzyme activity, and the characteristic protein expression of pyroptosis (Caspase-1, GSDMD-N, IL-1β, and IL-18), ultimately ameliorating BaP-induced pyroptotic damage by inhibiting the ROS/NF-κB/NLRP3/Caspase-1 signaling pathway in HL-7702 cells (Figure 8).

Gasdermin-D (GSDMD), a member of the Gasdermin family, is the common substrate of all inflammatory Caspase (Caspase-1/4/5/11) enzymes and is also the direct executor of pyroptosis in human and murine animals [31]. When pyroptosis occurs, GSDMD is cleaved by activated Caspase-1/4/5/11 into the N-terminal domain (GSDMD-N) with lipophilicity and pore-forming activity, and C-terminal domain (GSDMD-C) with hydrophilicity. GSDMD-N then selectively combined with the inner membrane of the cell membrane and oligomerized to form membrane pores with a diameter ranging from 10 to 20 nm, which caused the cell membrane permeability disorder and eventually led to cell swelling and rupture, resulting in pyroptotic death. At the same time, GSDMD-N could also promote the spillage of inflammatory factors IL-1β and IL-18 in large quantities, and stimulate a strong inflammatory reaction [32,33]. This work showed that Iso notably weakened the stimulatory effect of BaP on the oligomerization of GSDMD-N, the formation of cells membrane pores, the increase of the permeability of cell membrane, and the electrical conductivity, as well as the level of IL-1β and IL-18 (Figure 2 and Figure 7), indicating that Iso could suppress the BaP-induced pyroptotic hepatocytes damage.

According to different inflammatory Caspase and external stimuli, pyroptosis can be divided into the classic pathway depending on Caspase-1 and the non-canonical pathway depending on Caspase-4/5/11 [34]. The classic pyroptosis pathway is that Caspase-1 specifically cleaves and activates GSDMD to form GSDMD-N and GSDMD-C, and GSDMD-N further forms membrane pores on the cell membrane through oligomerization, which eventually leads to cell death. Meanwhile, Caspase-1 shears ProIL-1β and Pro-IL-18 into IL-1β and IL-18 overflowed through membrane pores. It has been reported that arsenic trioxide (As_2_O_3_) resulted in the nonalcoholic fatty liver disease/nonalcoholic steatohepatitis by inducing the classic pyroptosis pathway in HepG2 cells, accompanied by up-regulating the protein expressions of Caspase-1 and IL-1β [14]. As a Caspase inhibitor, Z-VAD-FMK alleviated HUVECs cells pyroptosis induced by CdCl_2_ [15]. To prove the genre of pyroptosis induced by BaP, Z-VAD-FMK was used to treat cells together with BaP. The results revealed that BaP resulted in the Caspase-1-dependent pyroptotic hepatocyte damage in HL-7702 cells (Figure 3). Consistent with the effect of Z-VAD-FMK, Iso decreased the Caspase-1 enzyme activity and the protein expressions of Caspase-1, GSDMD-N, IL-1β, and IL-18, demonstrating that Iso could inhibit the Caspase-1-dependent pyroptotic hepatocyte damage caused by BaP in HL-7702 cells (Figure 7).

Inflammasome is a type of polyprotein complex, including NLRP3, AIM2, Pyrin, NLRC4, and NLRP1. After the formation of the NLRP3 inflammasome with NLRP3, ASC and Pro-Caspase-1, the Pro-Caspase-1 will self-shear to form active Caspase-1, subsequently cleave and activate GSDMD to form GSDMD-N, and finally initiate pyroptosis [35,36]. MCC950 is a specific inhibitor of NLRP3. It was found that the addition of MCC950 could alleviate the pyroptosis induced by Mesoporous silica nanoparticles [37]. In this study, we also found that the MCC950 or Iso co-treatment with BaP could obviously weakened the activity of the NLRP3 inflammasome and the characteristic protein expressions of pyroptosis induced by BaP (Figure 4 and Figure 7), indicating that Iso could constrain the activation of the NLRP3 inflammatory response and the degree of damage of pyroptotic hepatocytes.

NF-κΒ is a nuclear transcription factor in the cells. It participates in the inflammatory response and regulates the expressions of inflammasomes [38]. It was found that the inhibition of NF-κΒ and NLRP3 inflammasome in macrophages could ameliorate colitis of mice [39]. The inhibitor of NF-κΒ (PDTC) was used to analyze whether or not NF-κΒ was involved in the activation of the NLRP3 inflammasome and pyroptosis reaction with cells. Iso and PDTC had similar effects, and both inhibited the nuclear localization of the p65 subunit of NF-κB, the activation of the NLRP3 inflammasome, and then initiating the Caspase-1-dependent pyroptotic hepatocyte damage induced by BaP (Figure 5 and Figure 7). Moreover, NF-κΒ did promote the activation of the NLRP3 inflammasome, and subsequently initiated pyroptotic hepatocyte damage.

BaP exposure would generate a large amount of ROS [40]. ROS is a trigger for the activation of the NLRP3 inflammasome, [21] and mtROS also activates the NLRP3 inflammasome through the NF-κΒ pathway [20,41,42]. ROS stimulated by Nicotine [43] and chronic ethanol [44] could induce the activation of the NLRP3 inflammasome and Caspase-1, and ultimately result in cells pyroptotic death. Chen’s research proved that pre-treatment with Mito-TEMPO (mtROS scavenger) or NAC (total ROS scavenger) suppressed Cd-induced activation of NLRP3 and pyroptotic cell death [15]. In the same way, our results revealed that ROS was a main contributor to BaP-induced pyroptosis and the activation of NF-κB and NLRP3 was confirmed by the fact that both ROS scavenger (NAC and Mito-TEMPO) and Iso could strikingly decrease the level of ROS and mtROS, and ameliorate the BaP-induced pyroptotic hepatocyte damage in HL-7702 cells through the inhibition of the ROS/NF-κB/NLRP3/Caspase-1 signaling pathway (Figure 6, Figure 7 and Figure 8).

## 5. Conclusions

In summary, Iso was proven to ameliorate BaP-induced pyroptotic hepatocytes damage in HL-7702 cells by suppressing the ROS/NF-κΒ/NLRP3/Caspase-1 signaling pathway, which will contribute to a theoretical basis for the hepatoprotective effect of Iso and the full revealment of the toxic effect and the control of BaP.

## Figures and Tables

**Figure 1 antioxidants-10-01275-f001:**
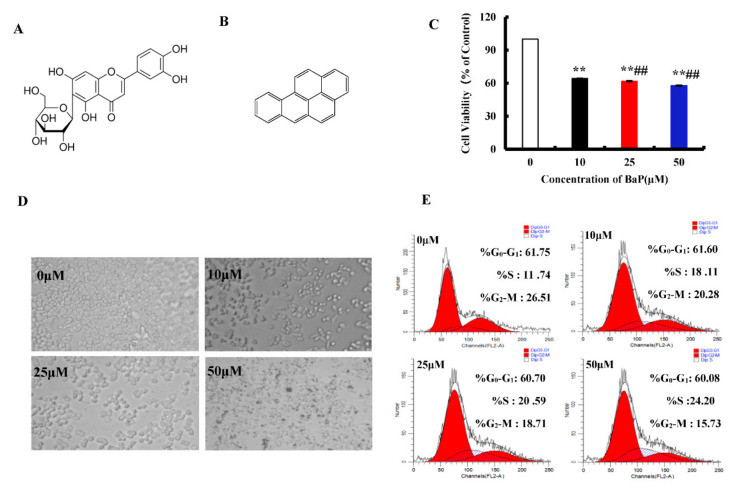
Effects of BaP on the proliferation of HL-7702 cells. Chemical structure of Iso (**A**) and BaP (**B**). Cells were incubated with BaP at different concentrations for 24 h and then processed for MTT assay. ** *p* < 0.01 compared with control, ## *p* < 0.01 compared with 10 µM BaP (**C**). Effect of BaP on cell morphology, which was observed with an optic microscope (**D**). Representative histograms of DNA content and cell cycle phases (G_0_-G_1_, S and G_2_-M) in BaP treated cells (**E**).

**Figure 2 antioxidants-10-01275-f002:**
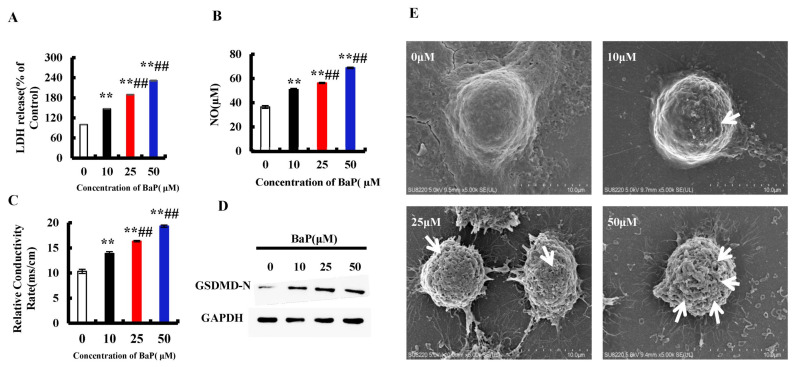
BaP induced pyroptotic hepatocyte damage in HL-7702 cells. LDH (**A**), NO (**B**), and electrical conductivity (**C**) were detected ** *p* < 0.01 compared with control, ## *p* < 0.01 compared with 10 µM BaP. Western blot image of protein expression acquired by Chemical XRS Imaging System (**D**). Scanning electron microscope observation of membrane pores; white arrows indicate the membrane pore (**E**).

**Figure 3 antioxidants-10-01275-f003:**
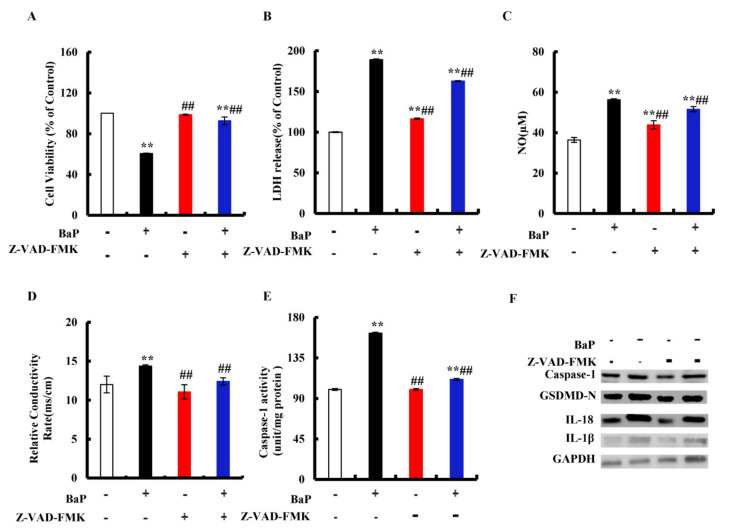
Effects of Caspase-1 inhibitor (Z-VAD-FMK, 20 µM) on pyroptotic hepatocyte damage. Cell viability (**A**), LDH (**B**), NO (**C**), electrical conductivity (**D**), and Caspase-1 enzyme activity (**E**) were detected. ** *p* < 0.01 compared with control, ## *p* < 0.01 compared with 25 µM BaP. Western blot image of protein expression acquired by Chemical XRS Imaging System (**F**).

**Figure 4 antioxidants-10-01275-f004:**
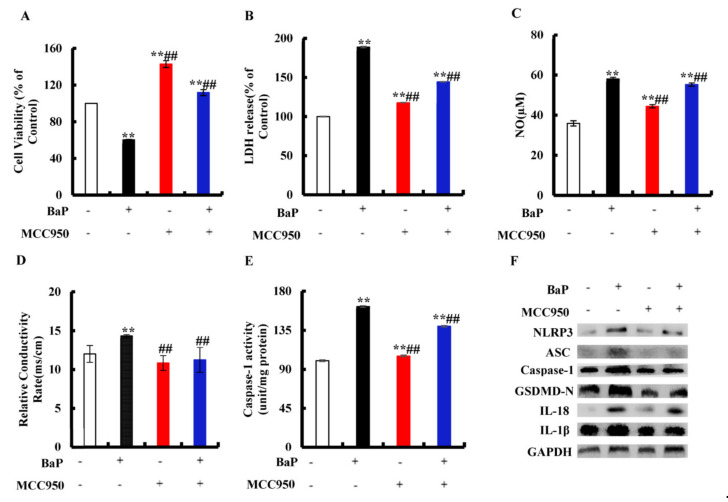
Effects of NLRP3 inhibitor (MCC950, 10µM) on pyroptotic hepatocytes damage induced by BaP (25µM). Cell viability (**A**), LDH (**B**), NO (**C**), electrical conductivity (**D**), and Caspase-1 enzyme activity (**E**) were detected. ** *p* < 0.01 compared with control, ## *p* < 0.01 compared with 25 µM BaP. Western blot image of protein expression acquired by Chemical XRS Imaging System (**F**).

**Figure 5 antioxidants-10-01275-f005:**
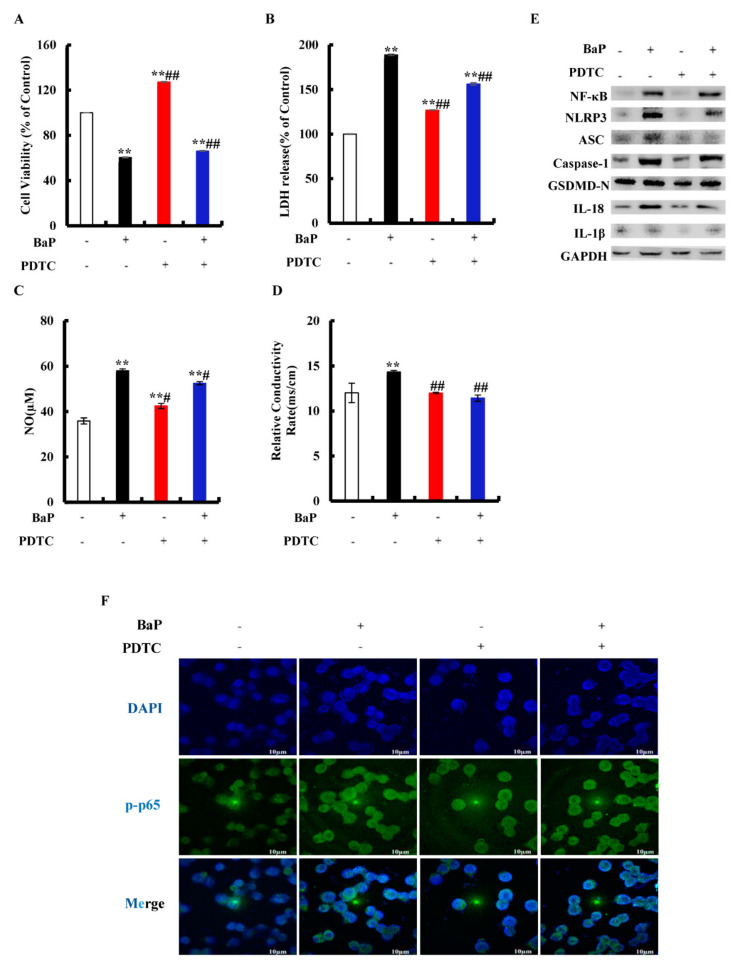
Effects of NF-κB inhibitor (PDTC, 50 µM) on pyroptotic hepatocyte damage induced by BaP (25 µM). Cell viability (**A**), LDH (**B**), NO (**C**), and electrical conductivity (**D**) were detected. ** *p* < 0.01 compared with control, ## *p* < 0.01 compared with 25 µM BaP. Western blot image of protein expression acquired by Chemical XRS Imaging System (**E**). Nuclear translocation of phosphorylated p65 was observed by immunofluorescent labeling with DAPI (blue) and anti-phosphorylated p65 (green) (**F**).

**Figure 6 antioxidants-10-01275-f006:**
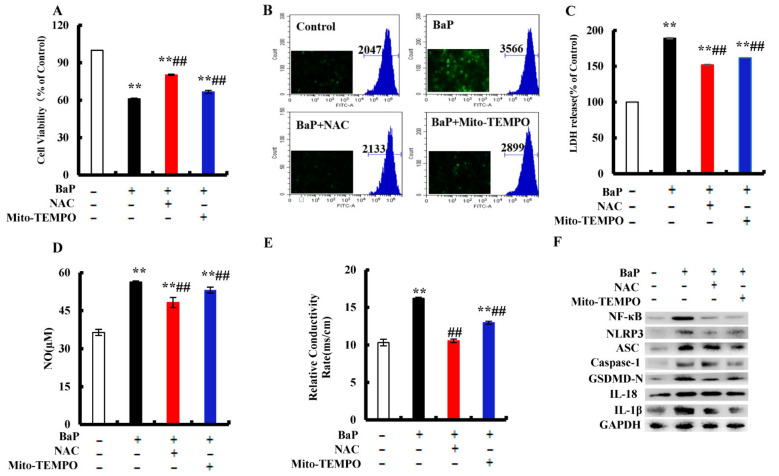
Effects of ROS inhibitor (NAC, 2 mM) and mtROS inhibitor (Mito-TEMPO, 50 µM) on BaP-induced pyroptotic hepatocyte damage. Cell viability (**A**), LDH (**C**), NO (**D**), and electrical conductivity (**E**) were detected. ** *p* < 0.01 compared with control, ## *p* < 0.01 compared with 25 µM BaP. ROS were examined by Flow cytometer and Fluorescence microscope (**B**). Western blot image of protein expression acquired by Chemical XRS Imaging System (**F**).

**Figure 7 antioxidants-10-01275-f007:**
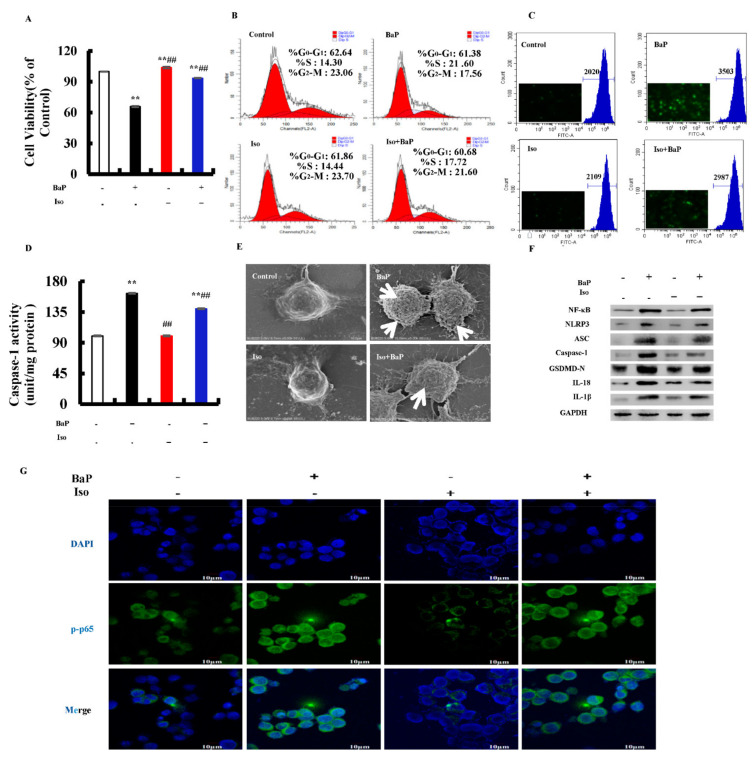
Effects of Iso (5 µM) on the pyroptotic hepatocyte damage induced by BaP (25 µM). Cell viability (**A**), cell cycle (**B**), ROS levels (**C**), and Caspase-1 enzyme activity (**D**) were detected. ** *p* < 0.01 compared with control, ## *p* < 0.01 compared with 25 µM BaP. Scanning electron microscope observation of membrane pores; white arrows indicate the membrane pore (**E**). Western blot image of protein expression acquired by Chemical XRS Imaging System (**F**). Nuclear translocation of phosphorylated p65 was observed by immunofluorescent labeling with DAPI (blue) and anti-phosphorylated p65 (green) (**G**).

**Figure 8 antioxidants-10-01275-f008:**
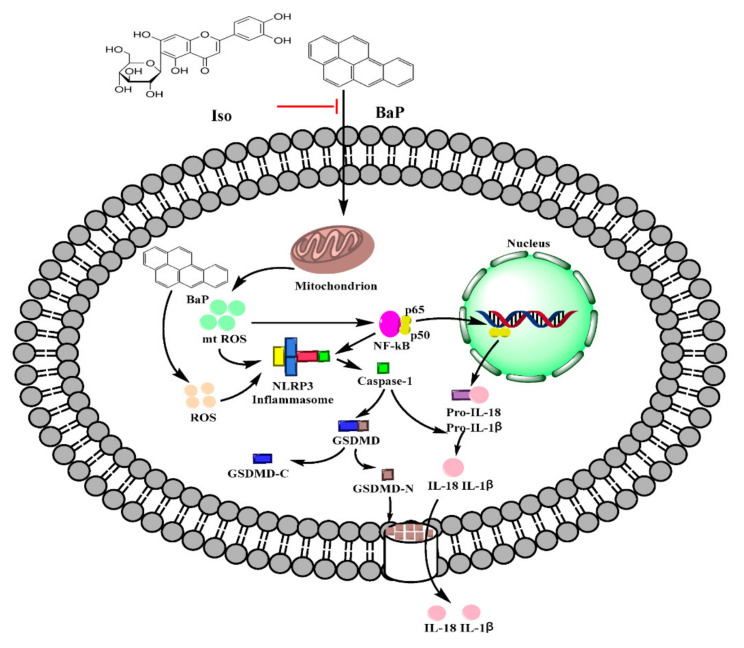
Possible molecular mechanism of Iso on the pyroptotic hepatocyte damage induced by BaP in HL-7702 liver cells.

## Data Availability

Data is contained within the article.

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
