# Peer review of "Isoorientin Attenuated the Pyroptotic Hepatocyte Damage Induced by Benzo[a]pyrene via ROS/NF-κB/NLRP3/Caspase-1 Signaling Pathway"

_antioxidants, 2021, doi:10.3390/antiox10081275_

Round 1

Reviewer 1 Report

The authors investigate the effect of antioxidant Isoorientin on Benzo[a]pyrene-induced pyroptosis, providing interesting results about the molecular mechanisms involved in the process. Overall, the study is well planned and therefore it is appropriate to be published on Antioxidants. I have few suggestions to improve the quality of the manuscript:

  • Line 32: “pro-inflammatory cytokines” should be used instead of the too general “inflammatory cytokines”
  • Graphs are sometimes too small, and therefore the text within the figure is difficult to read. See for example Figure 1 E.

Author Response

Dear Reviewer:

Thank you very much for the valuable comments regarding our manuscript “Isoorientin attenuated the pyroptotic hepatocytes damage induced by benzo[a]pyrene via ROS/NF-κB/NLRP3/Caspase-1 signaling pathway” (Antioxidants-1327809). We have modified text according to your suggestions, and the major revisions were highlighted in the revised manuscript using Track Changes. The point-by-point answers to the comments and suggestions were listed as follows:

The authors investigate the effect of antioxidant Isoorientin on Benzo[a]pyrene-induced pyroptosis, providing interesting results about the molecular mechanisms involved in the process. Overall, the study is well planned and therefore it is appropriate to be published on Antioxidants. I have few suggestions to improve the quality of the manuscript:

1.Line 32: “pro-inflammatory cytokines” should be used instead of the too general “inflammatory cytokines”

Response: Yes, we accept the reviewer’s suggestion, and have corrected it in the revised manuscript (please see the line 32 on the page 1).

2.Graphs are sometimes too small, and therefore the text within the figure is difficult to read. See for example Figure 1 E.

Response: Yes, we accept this comment, have redrawn all graphs in the revised manuscript.

Finally, we appreciate very much for your valuable suggestions and comments, and express our sincere thanks again.

Best Wishes.

Hao Li       lihr13@snnu.edu.cn

Li Yuan      yuanli112086@snnu.edu.cn   

Reviewer 2 Report

Li et al., Isoorientin attenuated the pyroptotic hepatocytes damage in- 2

duced by benzo[a]pyrene via ROS/NF-κB/NLRP3/Caspase-1

signaling pathway

In this study, Li et al found that isoorientin (Iso) attenuated the BaP (benzo[a]pyrene)-induced pyroptotic hepatocytes damage through inhibiting ROS/NF-κB/NLRP3/ Caspase-1 signaling pathway, which provides a new perspective and strategy to prevent liver injury induced by BaP. This study is appropriately designed and conducted. Nevertheless, there are some issues to be addressed:

Major comments

  1. The authors performed meticulous experiments to prove that Bap induced hepatocyte injury via the ROS/NF-κB/NLRP3/Caspase-1 signaling pathway, which are shown in Results 3.1-3.6. However, only a smaller part of experiments were performed to survey the influences of Iso on ROS/NF-κB/Caspase-1. It is suggested that the influence of Iso on NLRP3 inflammasome should be surveyed as well.

Minor comments

  1. Results, 3.1 Effect of BaP on the proliferation of HL-7702 cells: The authors found that BaP significantly prolonged the S-phase then jumped into the conclusion that BaP inhibited the growth of HL-7702 cells by blocking the S-phase in the cell cycle. Is it possible that BaP might on the contrary increased the synthesis of cells, because it prolonged the S-phase? Please clarify this point.
  2. Results, 3.1 Effect of BaP on the proliferation of HL-7702 cells: The authors claimed that BaP induced the death of HL-7702 cells 173 in a dose-dependent manner at three concentrations of 10 µM, 25 µM and 50 µM. However, there was no significant difference between cell viabilities of 25 µM and 50 µM groups (Fig. 1C).
  3. Some typo (or grammar) errors are to be corrected. Such as the end of the line 213:…..was been… and line 217: ……were been……

Author Response

Dear Reviewer:

Thank you very much for the valuable comments regarding our manuscript “Isoorientin attenuated the pyroptotic hepatocytes damage induced by benzo[a]pyrene via ROS/NF-κB/NLRP3/Caspase-1 signaling pathway” (Antioxidants-1327809). We have modified text according to your suggestions, and the major revisions were highlighted in the revised manuscript using Track Changes. The point-by-point answers to the comments and suggestions were listed as follows:

Li et al., Isoorientin attenuated the pyroptotic hepatocytes damage induced by benzo[a]pyrene via ROS/NF-κB/NLRP3/Caspase-1 signaling pathway. In this study, Li et al found that isoorientin (Iso) attenuated the BaP (benzo[a]pyrene)-induced pyroptotic hepatocytes damage through inhibiting ROS/NF-κB/NLRP3/ Caspase-1 signaling pathway, which provides a new perspective and strategy to prevent liver injury induced by BaP. This study is appropriately designed and conducted. Nevertheless, there are some issues to be addressed:

Major comments

  1. The authors performed meticulous experiments to prove that Bap induced hepatocyte injury via the ROS/NF-κB/NLRP3/Caspase-1 signaling pathway, which are shown in Results 3.1-3.6. However, only a smaller part of experiments were performed to survey the influences of Iso on ROS/NF-κB/Caspase-1. It is suggested that the influence of Iso on NLRP3 inflammasome should be surveyed as well.

Response: Thanks the reviewer’ comments. NLRP3 inflammasome is a multiprotein complex, formed by NLRP3, ASC and Pro-Caspase-1. Results 3.7 showed that Iso could reduce the increase of Caspase-1 enzyme activity, protein expression of NLRP3 and ASC, which indicated that Iso could inhibit the activation of NLRP3 inflammasome. Additionally, by detecting cell viability, the pores on the cell membrane, ROS levels, the nuclear localization of NF-κB, the activation of NLRP3 inflammasome and the characteristic protein expressions of pyroptosis, this study demonstrated that the natural flavonoid Iso attentuated BaP-induced pyroptotic hepatocytes damage through ROS/NF-κB/NLRP3/Caspase-1 signaling pathway in HL-7702 liver cells.

Minor comments

  1. Results, 3.1 Effect of BaP on the proliferation of HL-7702 cells: The authors found that BaP significantly prolonged the S-phase then jumped into the conclusion that BaP inhibited the growth of HL-7702 cells by blocking the S-phase in the cell cycle. Is it possible that BaP might on the contrary increased the synthesis of cells, because it prolonged the S-phase? Please clarify this point.

Response: Yes, we agree with the reviewer’ comments. Cell cycle is divided into interphase and mitotic phase, and interphase is divided into G1 phase, S phase and G2 phase. The S phase is the period of DNA replication, namely synthesis of cells. BaP prevented DNA replication and reduced the proportion of cells in the S phase, cell division will be stopped at the mitotic phase.

  1. Results, 3.1 Effect of BaP on the proliferation of HL-7702 cells: The authors claimed that BaP induced the death of HL-7702 cells 173 in a dose-dependent manner at three concentrations of 10 µM, 25 µM and 50 µM. However, there was no significant difference between cell viabilities of 25 µM and 50 µM groups (Fig. 1C).

Response: We are so sorry for the misunderstand. The viability of cells in 25µM BaP group and 50µM BaP group were 61.767±0.037% and 57.853±0.034%, respectively. There were significant (p<0.01) differences among these groups. We have corrected in the revised manuscript.

  1. Some typo (or grammar) errors are to be corrected. Such as the end of the line 213:…..was been… and line 217: ……were been……

Response: We followed the reviewer’s suggestion, and have corrected in the revised manuscript.

Finally, we appreciate very much for your valuable suggestions and comments, and express our sincere thanks again.

Best Wishes.

Hao Li       lihr13@snnu.edu.cn

Li Yuan      yuanli112086@snnu.edu.cn